# Order matters: How covert value updating during sequential option sampling shapes economic preference

**Chen Hu** [1,2] *, **Philippe Domenech** [3,4,5], **Mathias Pessiglione** [1,2] *

**1** Motivation, Brain & Behavior (MBB) team, Paris Brain Institute, Pitié-Salpêtrière Hospital, Paris, France, **2** Inserm Unit 1127, CNRS Unit 7225, Sorbonne Université, Paris, France, **3** Neurophysiology of Repetitive Behavior (NerB) team, Paris Brain Institute, Pitié-Salpêtrière Hospital, Paris, France, **4** Assistance Publique —Hôpitaux de Paris, GHU Henri Mondor, DMU Psychiatrie, Créteil, France, **5** Université Paris-Est, Créteil, France

* chenhu.judy@gmail.com (CH); mathias.pessiglione@gmail.com (MP)

**Data Availability Statement:** All data files and scripts for analysis are available on the Github: https://github.com/chenhucogneuro/ploscompbio_sequentialdecisionmaking/tree/master/SDM_DATA_openaccess. The VBA toolbox for the

## Abstract

Standard neuroeconomic decision theory assumes that choice is based on a value comparison process, independent from how information about alternative options is collected. Here, we investigate the opposite intuition that preferences are dynamically shaped as options are sampled, through iterative covert pairwise comparisons. Our model builds on two lines of research, one suggesting that a natural frame of comparison for the brain is between default and alternative options, the other suggesting that comparisons spread preferences between options. We therefore assumed that during sequential option sampling, people would 1) covertly compare every new alternative to the current best and 2) update their values such that the winning (losing) option receives a positive (negative) bonus. We confronted this "covert pairwise comparison" model to models derived from standard decision theory and from known memory effects. Our model provided the best account of human choice behavior in a novel task where participants (n = 92 in total) had to browse through a sequence of items (food, music or movie) of variable length and ultimately select their favorite option. Consistently, the order of option presentation, which was manipulated by design, had a significant influence on the eventual choice: the best option was more likely to be chosen when it came earlier in the sequence, because it won more covert comparisons (hence a greater total bonus). Our study provides a mechanistic understanding of how the option sampling process shapes economic preference, which should be integrated into decision theory.

## Author summary

According to standard views in neuroeconomics, choice is a two-step process, with first the valuation of alternative options and then the comparison of subjective value estimates. Our working hypothesis is, on the contrary, that the comparison process begins during the sequential sampling of alternative options. To capture this idea, we developed a computational model, in which every new alternative is compared with the current best,

analysis is available from: https://mbb-team.github.io/VBA-toolbox/.

**Funding:** The project was done during CH's PhD which was jointly funded by a French regional scholarship DIM cerveau et pensée (https://www.dimcerveaupensee.fr/) and the ICM (https://icm-institute.org/en/). The study was funded by the ICM (https://icm-institute.org/en/). The funders had no role in study design, data collection and analysis, decision to publish, or preparation of the manuscript.

**Competing interests:** The authors have declared that no competing interests exist.

so as to better contrast their values. This model provided the best account of choice behavior exhibited by participants (n = 92 in total) performing three variants of a novel multi-alternative decision task. Thus, our findings unravel a covert pairwise comparison process, occurring while participants collect information about alternative options, before they are requested to make their choice. They also provide explanations about when this covert process is implemented (when resampling is too costly), why it is implemented (to better discriminate the best options) and how it can bias decisions (because it favors first-encountered valuable options).

## Introduction

In everyday modern life, people often make choices between multiple options they can browse through, for instance when shopping for groceries, on the internet or at the supermarket. Even when options are readily available, located next to each other, they cannot be attended all at once. Sampling options is therefore a sequential process that unfolds across time, breaking equity between options by assigning them a serial position. This sequential allocation of attention is typically neglected in standard decision theory [1–3], for which economic choice is construed as the selection of the option maximizing expected utility. Thus, according to standard decision theory, the way information about alternative options is collected, in particular their position in the sampling sequence, should not have an impact on the eventual choice. Our working hypothesis is, on the contrary, that option sampling is not just passive information gathering, but an active process that covertly updates the utility function on which the eventual choice is based.

Several choice biases induced by sequential option sampling have already been reported. One notable bias is the so-called primacy effect, i.e. the fact that the first encountered option is more likely to be selected in a variety of choices, from picking a wine to joining a social group [4–7]. The primacy effect on choice has been classically interpreted as a by-product of memory functional properties, the first item in a series being more easily retrieved [8]. The reverse serial-position effect, namely the recency effect (i.e., the tendency to select the last encountered option) has been observed in choice as well [9] and again interpreted as a memory bias, in this case favoring the last item of the series [10]. The primacy and recency effects show that the position of options in a sequence can impact the final choice, but provides no evidence that decision making is already engaged during option sampling, as they are independent from preferences between options.

The aim of the present study was to test the intuition that people start comparing options as soon as they engage in the sequential sampling process. The purpose of these covert comparisons would be to provide a better contrast between options, such that preferences are easier to derive when the final choice must be made. For the sake of simplicity, and because it might reflect a true cognitive limitation of the human brain, we assumed that people compare options two-by-two. A natural frame for these pairwise comparisons would oppose the current best (or default) option to any new alternative. This decision frame has already been extensively investigated in foraging theory [11], for which the two key options in ecological choices are staying on the same patch versus switching to a new patch [12,13]. More recently, we provided evidence that a default versus alternative representational frame is also used when deciding between two goods [14], the default choice being defined by prior preferences (before further consideration of available options).

Thus, our hypothesis is that during naturalistic choice, where the number of options available is not known in advance but unveiled sequentially, people would compare at every step the current best to the newly sampled option. To investigate this hypothesis, we needed behavioral signatures of the covert pairwise comparison process. We reasoned that covert choice might spread preferences for compared options, as it has been shown with overt choice in research on cognitive dissonance [15–18]. The spread of alternatives has been shown in paradigms where participants first rate the likeability of a set of items, then items with similar ratings are paired and participants choose their favorite one within each pair, and last they rate every item in the list again. The classical observation is that the difference in likeability between chosen and unchosen items is greater in the last (post-choice) set of ratings than it was in the initial (pre-choice) set. Both behavioral manipulations and computational analyses have shown that the choice-induced spread of alternatives go beyond what could be expected from a statistical regression to the mean [19–21].

Instead of utility or preference, we employ here the term value to designate the subjective estimate of how good an outcome would be for the decision-maker, as is common in neuroeconomics [22,23]. If our hypothesis is correct, the covert pairwise comparison implemented at every step of option sampling should therefore enhance the subjective value assigned to the better option and decrease the subjective value attached to the worse option. The prediction is that the eventual (overt) choice distribution should reflect these (covert) subjective value updates. In addition, confidence in the choice, defined as the subjective belief that the (overtly) chosen option was indeed the best, should also reflect (covert) subjective value updates. This second prediction stems from previous reports that confidence increases with the distance in subjective value between chosen and unchosen options [24], as it does with the distance in stimulus strength for perceptual decisions [25]. Crucially, our model implies that the sequence in which options are sampled will have an impact on the final choice: if the best option is encountered earlier in the series, it will win more covert comparisons, which should further boost its subjective value and hence the likelihood that this option will be selected in the end, as well as the confidence that this option was indeed the best.

We compared our model to alternative models, derived from pure economic decision theory (softmax function of option values) or memory accounts (additive bias depending on serial position). We also tested more subtle alternatives, such as the progressive elimination of losing options in covert choices, instead of value updating. These models were compared on the basis of behavioral choice data collected in a novel task requiring participants to sample a set of 3 to 6 goods displayed on the screen, then selecting their favorite option in the presented set and, finally, reporting their confidence on a rating scale. In three successive experiments, we implemented variants of this design to better specify the context in which the covert comparison process might occur. Results show that our model provides the best account of choice data when going back to previously sampled options is made impossible or at least costly.

## Methods

### Ethics statement

The research was approved by the Ethics Committee for Biomedical Research ('Comité de Protection des Personnes') of the Pitié-Salpêtrière Hospital in Paris, France. All participants gave written informed consent before participating in the study. All studies were conducted following French rules and regulations.

### Participants

Three groups of participants (sample size: 30, 29, 33; gender: 18/12, 13/16, 21/12 female/male; age ± SEM: 24.3 ± 0.6, 24.8 ± 0.9, 25.1 ± 0.7, respectively) were recruited from a volunteer

database (the RISC, a local online cognitive science information relay platform for subject recruitment in the region of Paris, France) to participate in experiments 1 to 3. The recruitment criterion were healthy subjects between 20 to 40 years old, right-handed, no history or current treatment of neuropsychological disorders, and no particular dietary constraints.

The sample size was based on educated guess rather than formal power calculation, as we could not know in advance whether the effect of serial position would be present or not. A posteriori, given the observed size of the bonus parameter and its standard deviation across participants, a formal estimation indicated that a group of n = 20 participants would have been sufficient to test the effect with significance at 0.05 and power at 0.9.

Participants received a fixed gratification of 30 €, plus a randomly drawn item among all chosen options during their experiment, which was endowed to make sure that choice trials were incentive-compatible. To ensure that the payoff rule was plausible, participants were exposed to a large subset of potential rewards that were stored in the testing room. They were then informed about the random lottery implemented at the end of the task in order to determine their prize.

## Stimuli and apparatus

We used a total of 430 items in Exp 1 and 180 items in Exp 2 & 3 which were presented as standardized photos (400 x 300 pixels) on a computer screen. To avoid boredom, we selected items from different categories: processed food (sweet and salty snacks), unprocessed food (vegetables and fruits), films, music albums and readings (newspapers or magazines). The five categories were used in Exp 1, only three in Exp 2 and 3 (unprocessed food and reading categories were discarded).

The majority of food items were from the French INSEAD food database, the other items were created in the lab. Food products were photographed frontally over a black background, with some of the contents displayed in front of the packaging. Items from the music category (album cover with name of singer as it is popularized in the French market) and from the reading category (journal or magazine logo over a black background) were recycled from previous studies. Films were selected among the most popular ones in France during the past 10 years, according to an online film database, and the corresponding images were created in the lab from the French version of movie posters. Although we intended to present familiar items, it remained possible that a participant never heard of a particular film or music album. In this case, the participant was asked to evaluate its likeability on the basis of previous experience with items of a similar kind.

All experimental stimuli were presented with Matlab_R2017a (https://www.mathworks.com/) running Psychophysics Toolbox-3 [26] (http://psychtoolbox.org) and additional custom scripts. The experiment was run on a 20-inch touch screen Windows-based PC with a resolution of 1920 * 1080.

## Behavioral task

The tasks are not numbered in chronological order: Exp 3 (the control experiment) was in practice conducted between Exp 1 and Exp 2 (the test experiments). Instead, for the sake of readability, they are ordered following the difficulty of resampling the options, which was made impossible in Exp 1, costly in Exp 2 and free in Exp 3 (Fig 1).

In Exp 1, the five categories of stimuli were divided into five sessions. Each session was composed of a rating task and a choice task using items from the same category. In the rating task, participants were asked to assign likeability ratings to each of the 86 items. They were instructed to move a cursor along a visual analog scale on the computer touch screen. The

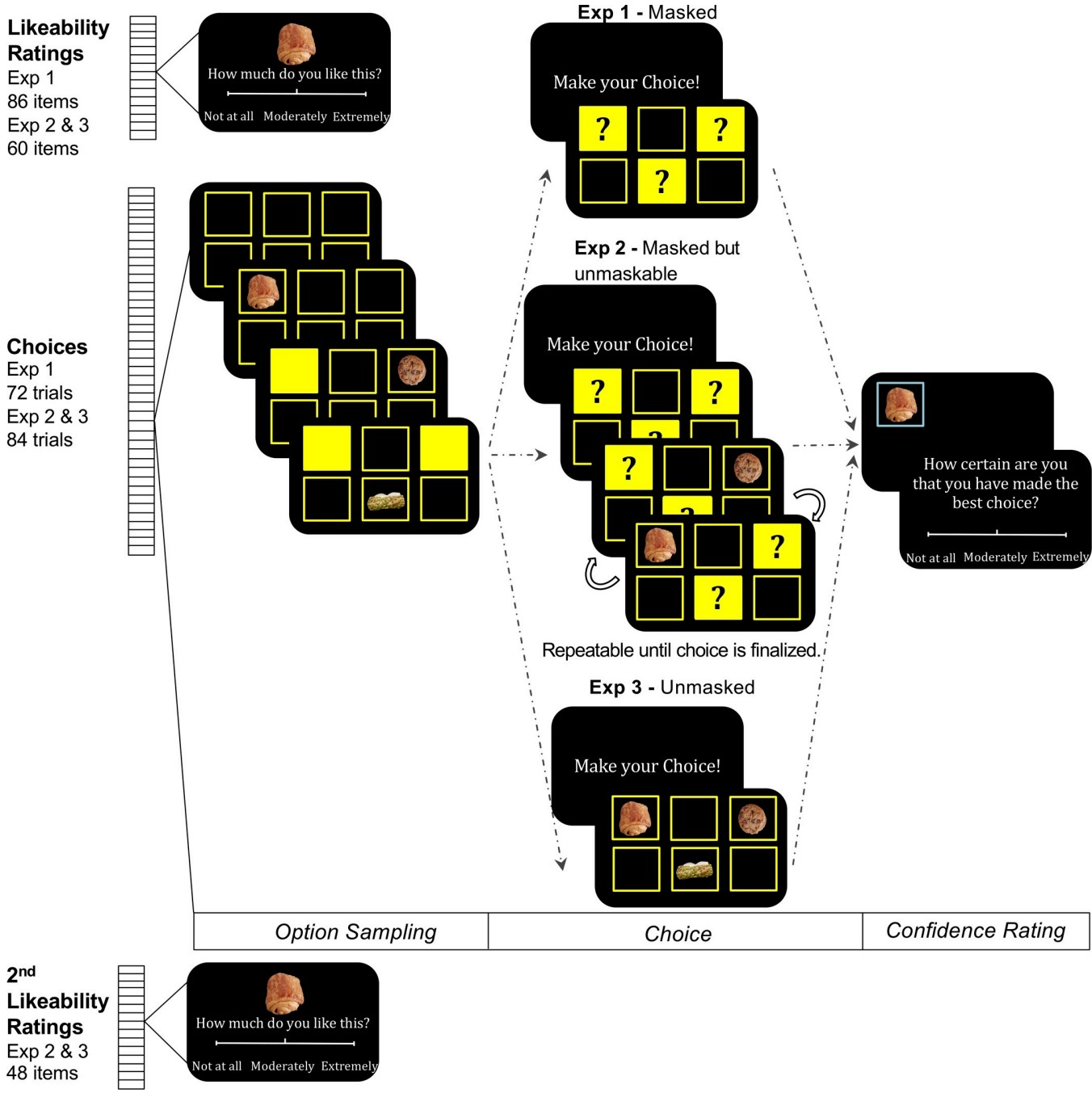

**Fig 1. Behavioral tasks.** Each task session used one category of items (food, music, film or magazine) and was divided into two stages. In the first stage, participants had to rate the likeability of items presented one by one on the screen, by placing a cursor on a visual analog scale. In the second stage, participants had to browse through a set of 3 to 6 options, by pressing the space bar on the keyboard, then click on their favorite option at the end of the sequence, and last, rate their confidence in their choice by placing a cursor on another visual scale. An additional third stage was included in Exp 2 and 3, with a second likeability rating task identical to the first one (presenting the same items). Beyond details regarding the number of items and categories, key differences between experiments regarded the choice phase: after being sampled, options remained masked (with no possibility of unmasking) in Exp 1, remained unmasked (with a time consuming possibility of resampling) in Exp 2, and were simply unmasked all together in Exp 3. Option sampling was self-paced, participants proceeding to the next option by pressing the space bar. At each step, a new option was revealed while the previous one was masked again. The location of the options appearing on screen was randomized but their identity was prearranged on the basis of likeability ratings (see methods). Choice was prompted by displaying question marks on masks (Exp 1 and 2) or by unmasking all options together (Exp 3). A feedback showing the chosen option alone was provided before the confidence rating.

scale was extended from '*not at all*' to '*extremely*' in response to the question '*how much do you like this item*'. The rated items were then ranked according to the subject-specific likeability ratings. Should two items have the same rating, we would compare the corresponding response time and assign a higher ranking to the one with a quicker response time. We then grouped the ranked items into choice trials in a way that balanced the position of the best item in the sequence (the highest initial rating compared to other items of a same trial). Note that rankings were only used to control the position of the best item in the sequence, never for data analysis or computational modeling, which instead were based on likeability ratings (i.e., cardinal and not ordinal values).

There were 72 trials per choice session. Choice trials began with a fixation cross (2 seconds), followed by six equally distanced empty squares (three on the upper half and three on the lower half of the screen). Then, the sequential sampling phase started. At each step of the sequence, a new option was displayed at a random location among the six placeholder squares. Participants sampled three to six options before they were asked to make a choice. The association between placeholder squares and options was fixed for the entire trial. Participants could sample the following options one-by-one in a self-paced manner. Each time when they pressed the space key, a new option showed up on the screen. Meanwhile, the previous one was masked by a filled square. The length of the sequence was not known to them before they reached the end of the trial. After sampling all options of a given trial, they were cued with a choice prompt ('make your choice'), followed by question marks placed on all filled squares masking the available options. Participants could then choose their favorite option by clicking on the corresponding square. Thereupon, a feedback screen appeared with the selected option unmasked for 1 second. At the end of the trial, participants were asked about their level of confidence. The question was '*how certain are you that you have made the best choice*' (in the sense of a choice corresponding to their own subjective preference). Confidence ratings were provided by placing a cursor on a visual analog scale graduated from '*not at all*' to '*extremely*'. Within a session, every item appeared in four different trials, each time paired with a different set of items. Choices, response times and confidence ratings were recorded for further analysis. At the end of the experiment, one out of all the trials was selected randomly and the chosen item at the selected trial was given to the participant, on top of the fixed payment.

Exp 2 had a similar setup as Exp 1, apart from the following changes. The critical change was that before finalizing their choice, participants had the opportunity to unmask any of the options, just by clicking on it. They could then confirm their selection by pressing the space bar, or revisit other options by clicking on other locations. Only one option would appear on the screen at a time: if one was unmasked, the previous one was masked again, such that pressing the space bar designated the option on screen as the final choice.

A more minor change was the adaptation of the visual presentation, such that instead of having one option revealed right after the fixation cross, participants would see six empty squares. Therefore, they needed to press the space key to reveal the first option, whose exposure time was thus better controlled. Other secondary changes pertain to the number of items and how they were organized into choice trials. Due to practical constraints related to testing time and potential fatigue (Exp 1 lasted about 3 hours), we reduced the number of options that participants had to rate and compare. Given that in Exp 1, different trial lengths did not result in different predictions or observations, we removed most of the trials with six options. Two such trials were kept per repetition, to avoid participants anticipating the end of the sequence, but were not included in the analyses. These catch trials were composed of items with the six lowest and six highest ratings, because extreme ratings might be bounded by the scale and thus less accurately estimated. We further removed the reading and unprocessed food categories, reducing the set of items to three categories only. Each of the items was presented six instead

of four times in the corresponding choice session, while ensuring that they were paired with different items at each repetition. In total, there were 60 items in a rating session, for a total of 84 choice trials (72 in data analysis) per session.

More importantly, we systematically controlled for the distance, in both likeability rating and serial position, between best and second-best options in a given trial. According to our predictions, preference reversals should be more easily observed in trials where the best and second-best options were close in value and far apart in the sequence (the best coming at the end, and the second-best at the beginning). Accordingly, the difference in value was kept under a maximum of three ranks, based on likeability ratings. We also avoided minimal differences in serial position in longer sequences (with 4 items and more), like consecutive display of best and second-best for trials including four or five options.

Exp 3 used the same set of options, with the same grouping scheme as introduced in Exp 2, resulting the same number of trials, and the same visual display, except for one critical change. During the choice phase, the masks were removed such that participants were able to see all the available options again before selecting their favorite option. This setup corresponds to a simultaneous presentation of alternative choice options as used in standard behavioral economics experiments. It can therefore be seen as a control for the effects observed in Exp 1 and 2.

Aside from adaptations in the choice task, we also included a second likeability rating session, after the choice session, in Exp 2 and 3. This second rating session was identical to the first one and served to assess the persistence of choice-induced effects on subjective value.

## Computational modeling

The central questions of the modeling work are whether and how sequential option sampling affects decision making.

We first formalized the null hypothesis (H0), which assumes that there is no update of value or pre-commitment to a choice during the option sampling process. Hence, at the time of choice, participants recall or reassign option values, which on average should be equal to their initial likeability ratings. These option values are then compared in a noisy selection process captured with a softmax function. In every trial, selection probabilities under H0 are therefore:

$$P_i = \frac{e^{\beta * V_i}}{\sum_j^o e^{\beta * V_j}} \tag{1}$$

Where $P_i$ is the probability of choosing item $i$ given its value $V_i$, and the value $V_j$ of all the options o displayed in the trial (including $i$). In model H0, all option values correspond to initial likeability ratings. There is only one free parameter to be fitted in this model, the inverse temperature $\beta$, which captures the weight of values (relative to pure noise) in choices. While selection probabilities are independent from option serial position in H0, the other models assume different mechanisms predicting an effect of the sequence order.

In H1 (primacy model), the sequence order affects option values through memory effects. For the sake of simplicity, we used a bias parameter that follows an inverse function of the serial position. According to H1, value updating at every step of option sampling is given by the following rule:

$$V_i(t_1) = V_i(t_0) + \frac{\lambda}{s_i} \tag{2}$$

where $V_i$ is the value of item i, $\lambda$ is the magnitude of the bias parameter, which is discounted by

the option serial position $S_i$. As in H0, selection probabilities are calculated with the softmax function (1), now using updated values $V_i(t_1)$ instead of initial ratings. The same model can be used to capture a primacy bias (if $\lambda > 0$) or a recency bias (if $\lambda < 0$).

To give the best chance to the hypothesis of a primacy bias, we also implemented other possible decay functions. In variant H1a, the bias was a power function of serial position:

$$V_i(t_1) = V_i(t_0) + \lambda^{s_i} \tag{3}$$

While in variant H1b, the bias was not additive but multiplicative:

$$V_i(t_1) = V_i(t_0) \times \left(1 + \frac{\lambda}{s_i}\right) \tag{4}$$

In H2 (bonus model), the sequence order affects option values through iterative covert pairwise comparison between the current best and the new option. At every step, the option with higher value receives a bonus and becomes the current best for the next comparison. Symmetrically, the option with lower value receives a negative bonus of the same magnitude. This symmetrical update ensures that values would keep the same mean throughout the sampling process. According to H2, the updating rule can be described as follows:

$$V_i(t_{ni}) = V_i(t_{ni-1}) + \delta \tag{5}$$

$$V_j(t_{nj}) = V_j(t_{nj-1}) - \delta \tag{6}$$

where values of options $i$ and $j$ are updated after a number of covert comparisons $t_{ni-1}$ and $t_{nj-1}$, respectively. Assuming that option $i$ has a higher value than option $j$ at the time of the new comparison, a bonus of size $\delta$ is added to option $i$ and subtracted from option $j$. Option $i$ is then set as the new default (current best option) for the next step.

In H1 and H2 models, just as in H0, selection probabilities for the choice phase are calculated with the softmax function (1), using updated values $V_i(t_{ni})$ instead of initial ratings $V_i(t_0)$. We also implemented a variant of H2 where the bonus $\delta$ is added to the first presented option, as if it was winning against a fictive option of null value (anything is better than nothing). This modified model was labeled H2.1, to make the link with H1, as it also incorporates a notion of primacy bias (applied to the first option only).

In H3 (pruning model), the sequence order affects selection probabilities by eliminating options instead of updating values, after each pairwise covert comparison. At every step, the probability of keeping the current best versus the new option is calculated as in (1), through a softmax function of their values, which remain equal to initial ratings. Therefore, the global likelihood that an option is eventually chosen can be estimated by the multiplication of all conditional probabilities along the corresponding branch in a probability tree diagram.

To identify differential predictions of models H0 to H3, we simulated datasets under Exp 1 design. Ratings were generated using a uniform distribution ranging from 0 to 100, slightly left-skewed towards 0, as was observed in a pilot study showing a subset of stimuli to an independent sample of subjects. The inverse temperature was extracted at first from a pilot experiment with similar designs, and then readjusted with the inverse temperature parameter fitted on choice data collected in Exp 1. The other parameters ($\lambda$ and $\delta$) were varied to examine their effects on choices. Trial-by-trial selection probabilities were generated for every simulated subject and averaged over 200 simulated datasets of 30 subjects. A critical difference across models was identified in the relationship between the probability of selecting the best option and its position in the sequence (Fig 2 and S1 Fig under the settings of Exp 1 and Exp 2–3, respectively). This relation was flat under H0, decreasing under H2, and increasing under H3.

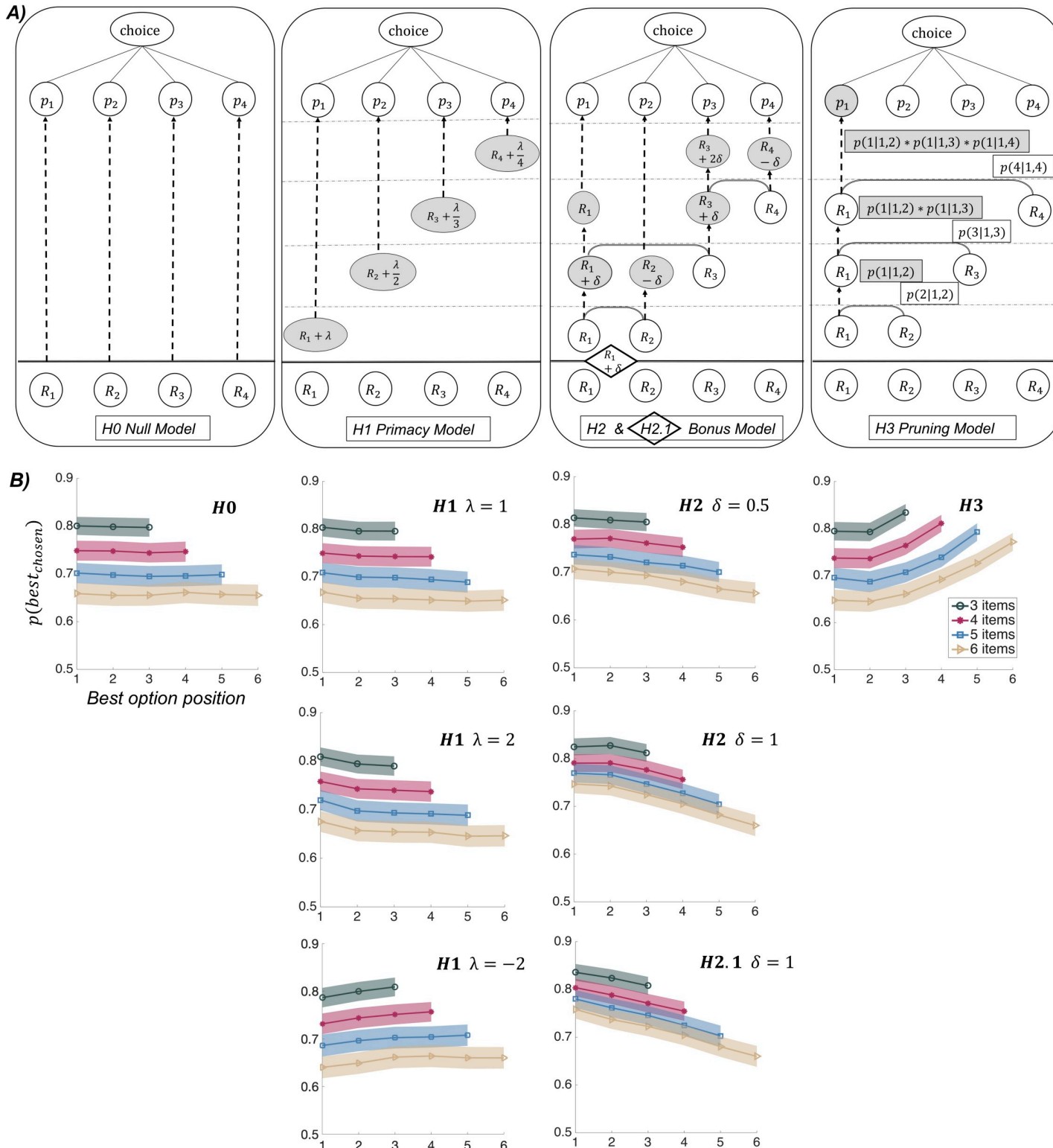

**Fig 2. Computational models and simulations of choice data.** A) Graphical depiction of computational models. For each model, a trial with four options, initially rated R1 to R4 (with R3 > R4 > R1 > R2), are illustrated. Progressive choice preparation should be read from bottom to top, horizontal dashed lines representing transitions between steps of sequential option sampling (each time a new option is revealed). In H0 (null model), option sampling is neutral, such that choice probabilities are given by a softmax function applied to initial ratings, as in standard decision theory, using one free parameter β (inverse temperature). In H1 and H2, option values are updated, either according to their serial position in the sequence (primacy model) or according to pairwise covert comparison between current best

and new options (bonus model). Grey circles represent updated values. In model H1, the positive parameter λ is used as a bias added to option value, whose magnitude decreases as the inverse function of the serial position in the sequence, in accordance with the notion of primacy effects. Note that the same model with negative λ would produce recency effects. In model H2, the positive parameter δ is a bonus added to the winning option, and subtracted from the losing option, in covert pairwise comparison. Winning and losing options are simply determined by their initial likeability ratings prior to the comparison. Model H2.1 applies the same logic to the first option (illustrated in diamond), which automatically benefits from the additive bonus δ as if it was winning the first comparison (against nothing), hence including a sort of primacy effect. In H3 (pruning model), covert pairwise comparisons are used to select the winning option, and eliminate the losing option, with a probability provided by a softmax function of their values. The winning option is the one compared to the new option in the next step. As the outcome of covert comparisons is probabilistic, the figure illustrates only one of many possible trees. B) Results of model simulations (under Exp 1 setup). Graphs show simulated choices as the probability of selecting the best option (y axis), depending on its serial position in the sampling sequence (x-axis), which was varied across trials. The same softmax function is used in all models to derive choice probabilities from option values, with an inverse temperature β = 0.081, corresponding to the posterior estimates obtained from fitting the best model (H2.1) to choice data collected in Exp 1. Each of the plots is an average over 200 simulated datasets of 30 subjects implementing the corresponding model, for various (color-coded) number of options. Shaded areas indicate the average inter-participant SEM, across all datasets. Values of λ and δ are indicated on the plots. Note that only H1 (with positive λ) and H2 (including H2.1) predict a decreased probability of choosing the best option when it is presented later in the sequence.

Regarding H1, it was decreasing with a positive bias (primacy model) and increasing with negative bias (recency model). Among the models with a negative relation, the shape was convex with model H1, concave with model H2, and linear with model H2.1, providing qualitative signatures. For a more quantitative assessment, we used Bayesian model comparison (see below).

## Statistical analyses

All analyses were run with Matlab_R2018b (www.mathworks.com). Effect sizes are provided as means and standard errors of the mean (SEM). Model-free analyses used linear regressions at the individual level, followed by two-tailed t-tests on regression coefficients, at the group level. Computational models were inverted using a Variational Bayes approach under the Laplace approximation. Model inversions were implemented using Matlab VBA-toolbox (available at http://mbb-team.github.io/VBA-toolbox/). This iterative algorithm provides a free-energy approximation for the model evidence, which represents a natural trade-off between model accuracy (goodness of fit) and complexity (degrees of freedom). Additionally, the algorithm provides an estimate of the posterior density over the free parameters, starting with Gaussian priors. Log model evidences were then taken to group-level random-effect analysis and compared using a Bayesian model selection (BMS) procedure. This procedure results in an exceedance probability (Ep) that measures how likely it is that a given model is more frequently implemented than the others in the population [27].

## Results

We recruited three different groups of participants ($n_1$ = 30, $n_2$ = 29, $n_3$ = 33) for the three experiments. There were no significant differences in gender, age or education level between groups. Three participants were excluded from the analyses (one in Exp 1 and two in Exp 2), either because they quitted the experiment before the end or gave invariant rating to every item. The analyses were conducted on the remaining participants ($n_1$ = 29, $n_2$ = 27, $n_3$ = 33).

## Model-free analysis

**Choice behavior.** To assess the qualitative predictions of our computational model, we tested the relationship between P(best), the probability of choosing the best option (defined as the option with highest initial rating), and the serial position of that best option. The linear regression was done separately for each trial length (number of options) in each participant. Regression coefficients were then averaged, to obtain one value per individual and experiment, and tested at the group level (Fig 3A). Regression coefficients were significantly negative in Exp 1 (b = - 0.015 ± 0.0052, t(28) = -2.86, p = 0.0079) and Exp 2 (b = - 0.020 ± 0.0084, t(26) = -

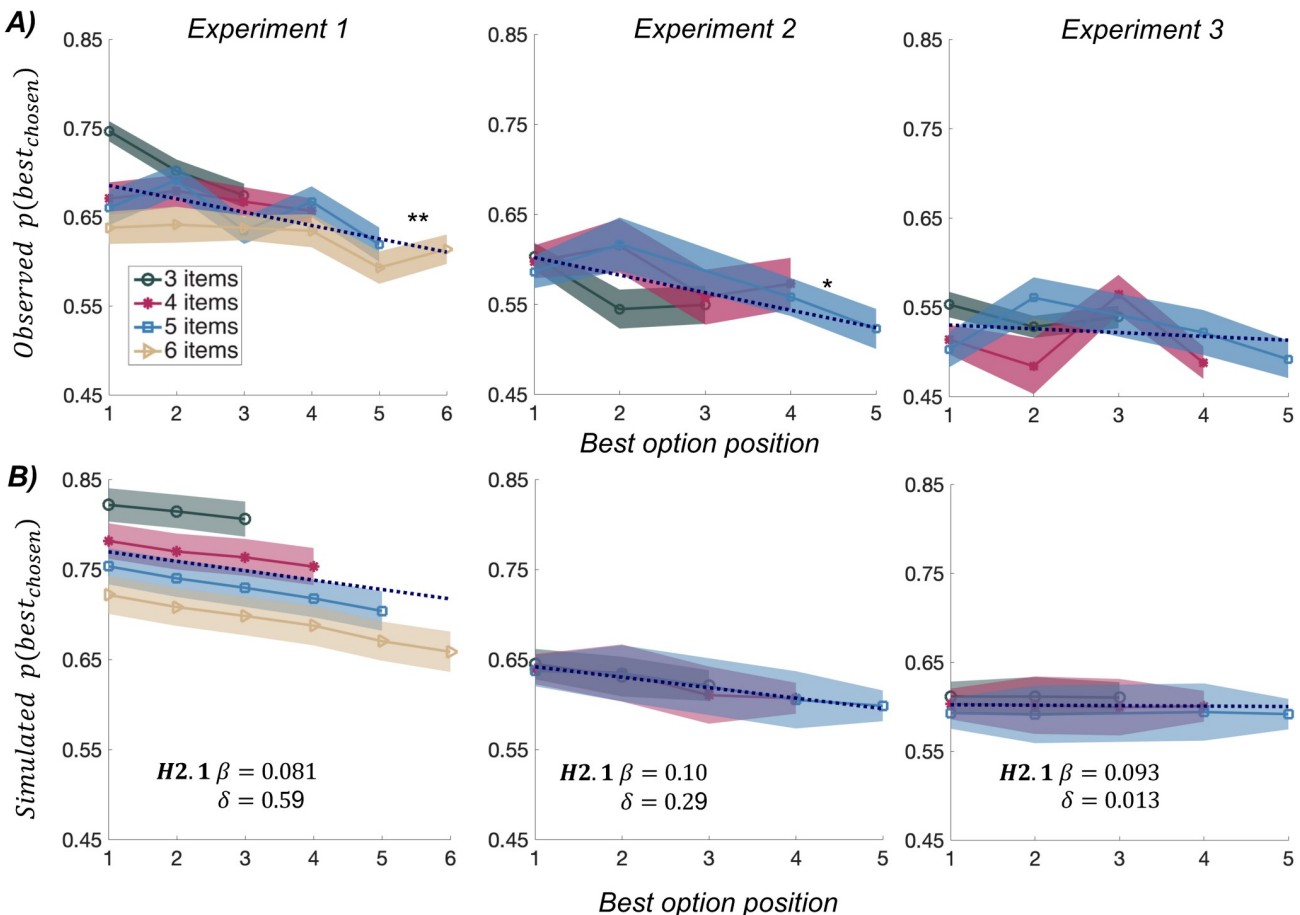

**Fig 3. Comparison of behavioral results to model simulations.** A) The upper graphs show the observed probability of choosing the best option, as a function of its serial position, for different (color-coded) number of options. Shaded areas indicate inter-participant SEM. Dotted lines show linear regression fit across all trials (with different numbers of options). Stars denote significance of t-test comparing regression slopes to zero. * p<0.05, ** p<0.01. B) The bottom graphs show the simulated probability of choosing the best option, as a function of its serial position. Choice behavior in each condition was simulated using the best-fitting model with the posterior means for free parameters (see values of the inverse temperature β and bonus δ indicated on the plots). Each of the plots is an average over 200 simulated datasets of 30 subjects implementing the corresponding model, for various (color-coded) number of options. Shaded areas indicate the average inter-participant SEM. across all datasets.

2.32, p = 0.028) but not in Exp 3 (b = - 0.0042 ± 0.0063, t(32) = - 0.67, p = 0.50). The conclusion is the same if we regress P(best) against the factor that was controlled in Exp 2 and 3, i.e. the relative serial position between best and second best (S2A Fig): regression coefficient was significantly negative in Exp 2 (b = - 0.010 ± 0.0042, t(26) = - 2.46, p = 0.021), but not in Exp 3 (b = -0.0011 ± 0.0029, t(32) = - 0.375, p = 0.71).

The negative linear link observed in Exp 1 and 2 is consistent with the predictions of the extended bonus model H2.1, while the flat relationship observed in Exp 3 corresponds to the null model H0 (see Fig 3B and S2B Fig). Exp 2 can be seen as similar to Exp 1, to the extent that participants rarely resampled the options once they were masked (only in 12% of trials), suggesting that they were sensitive to time costs. Thus, the order of option sampling was roughly that programmed in the sequence of option display. However, in Exp 3, participants had the liberty to resample the options at no cost (just by moving their eyes), meaning that we lost track of the sequential order, compared to what was programmed by design. Thus, Exp 3 can be taken as a control that the link between probability and position observed in Exp 1 and 2 was not artefactual.

Apart from the impact of serial position, there were clear differences in choice data between Exp 1 and the two others, related to changes in the design. An important change was that the difference in value was kept minimal between best and second-best options in Exp 2 and 3. The consequence was that, in the two last experiments, the best option was less frequently chosen (Exp 1: 0.66, Exp 2: 0.57, Exp 3: 0.52), while the second-best option was more frequently chosen (Exp 1: 0.21, Exp 2: 0.36, Exp 3: 0.37). For the other options, choice rate was lower, which explains why adding more (low-value) items in the sequence had a lesser effect in Exp 2 and 3.

A possible confound for the effect of serial position is exposure time, which has been found to bias decision-making in previous studies [28,29]. Here, we could only assess the impact of exposure time in Exp 2, because in Exp 1 the first option was shown before any key press, making it different from the next options, and in Exp 3 all options remained visible on screen, making exposure time difficult to estimate without tracking gaze fixation. When choice probability was regressed against a logistic model that contained both decision value (DV) and exposure time (ET), the two regressors were found to have a significant influence (DV: $b = 0.030 \pm 0.0019$, $t(26) = 15.68$, $p < 0.001$; ET: $b = 0.34 \pm 0.13$, $t(26) = 2.69$, $p = 0.012$). The dissociation between exposure time and option value is not easy to operate however, because participants tend to look longer at options they like better in the first place.

In any case, we checked that the putative impact of exposure time could not contribute to the observed effect of serial position. This would occur if the best option was looked longer when presented earlier in the sequence. It was not the case: when regressing exposure time against the best option serial position, just as we did for choice probability, the slopes were not different from zero ($b = 0.0026 \pm 0.012$, $t(26) = 0.22$, $p = 0.83$). Therefore, even if longer exposure time indeed contributed to a higher choice rate, this effect was orthogonal to our main research interest (i.e., the impact of serial position during sequential sampling).

**Confidence and response time.** If we define confidence as the selection probability of the chosen option (given by the softmax function of option values), then model H2.1 also predicts a lower confidence when the best option arrives later in the sequence, at least when that best option is chosen. To test this prediction, we estimated regression slopes between confidence ratings (in trials where best option was chosen) and best option serial position, as we did for choices. Consistent with the prediction, this regression slope was significantly negative ($b = -0.65 \pm 0.28$, $t(28) = -2.30$, $p = 0.029$) in Exp 1. We reasoned that the decrease in confidence (with best option serial position) should be paralleled by an increase in response time. Yet response time should be interpreted with caution in our experimental design, because it is confounded with target location, which affects the distance to touch screen. We nonetheless observed a trend (see S3 Fig) for a positive relationship between response time and best option serial position ($b = 0.012 \pm 0.0066$, $t(28) = 1.95$, $p = 0.061$). From these observations we may conclude that the putative covert pairwise comparison process not only increased the best option choice rate, but also increased confidence and decreased response time. However, these trends (decreased confidence and increased response time) were not observed in Exp 2, possibly because in this version of the task, participants could control their confidence level by resampling the options.

We therefore explored resampling behavior in more details, in Exp 2 where we could track it, although it remained rare (12% of trials on average). There was no clear spatial or temporal pattern: options presented at a given location on screen or at a given position in the sequence were not resampled more often than the others. However, compared to base rate (33, 25 and 20% for 3-, 4- and 5-item sequence), the best and second-best options were resampled significantly more often ($t(25) = 5.74$, $p < 0.001$ and $t(25) = 3.04$, $p = 0.0054$, respectively). We also compared confidence rating between trials with and without resampling: it was significantly

lower after resampling (from 77.53 ± 1.83 to 63.22 ± 2.61%, t(50) = -4.53, p <0.001). This result does not support the idea that resampling increased confidence, but it is hard to conclude on this point, because we have no access to their confidence level before participants start resampling. Thus, the data remain consistent with the possibility that participants resampled when their confidence was low, under the impression that they missed or forgot some information, with this resampling behavior not restoring their usual confidence level (even if it helped). However, no strong conclusion about resampling should be drawn here, because it was only observed in a limited subset of trials and subjects.

## Model-based analysis

**Within-trial effects.** The negative relationship between best-option choice rate and serial position is compatible not only with model H2.1 but also models H2 (with bonus only) and H1 (with primacy bias only). To further disentangle between these models, we implemented a Bayesian model comparison based on choice data.

In order to give the best chance to the primacy model, we first compared, after Exp 1, the different possible functions relating the primacy bias to the option serial position (i.e., H1, H1a and H1b). The results of Bayesian model selection suggested that H1 was the most plausible function (Ef = 0.76, Ep > 0.99). In the model space, we therefore included the null, primacy and bonus models (H0, H1, H2, and H2.1), but not the pruning model (H3), since it predicted a trend opposite to that observed in the data. For similar reasons, we bounded the prior of the bias parameter in the primacy model (H1) to be positive, because we did not observe any trend in the data that would reflect a recency bias.

To examine whether these models can be distinguished through Bayesian comparison, we conducted a model recovery analysis. Obviously, recovery success depends on computational parameters, since for instance, model H1 with a null $\lambda$ parameter is nothing but H0. What we intended to check is whether a winning model, with an exceedance probability over 0.95, could be confused with another in the model space, given the observed choice data. We therefore used the observed likeability ratings, and computational parameters fitted on our choice data, to simulate virtual groups of participants. Bayesian model comparison was then conducted on simulated data in the exact same manner as with observed data. Recovery rate was good (see S4 Fig), except that H0 was winning when H1 was simulated. This is likely related to the fact that the bias parameter $\lambda$ of model H1 was too small in our participants, making this model similar to model H0, which has the advantage of having one parameter less. Note that the impact of $\lambda$ diminishes with the number of items, while the impact of $\delta$ is cumulative. The critical point was that when H2.1 was winning, the simulated model was H2.1 in a majority of cases and H2 in the others, thus validating the hypothesis of a covert pairwise comparison process.

The results of group-level random-effect Bayesian model selection are reported separately for the three experiments (Fig 4). Model H2.1 (bonus + primacy model) was identified as the best model in Exp 1 (Ef = 0.63, Ep = 0.94) and Exp 2 (Ef = 0.76, Ep > 0.99). Although exceedance probability did not reach 0.95 in Exp 1, we note that its only competitor with non-zero exceedance probability was H2, meaning that in any case the hypothesis of covert pairwise comparison was the most plausible. It is also worth noting that, in Exp 2, model H2.1 won the comparison whether based on just the imposed sequential sampling steps or on all the actual sampling steps, including when participants took the opportunity to resample options after the sequential presentation. In Exp 3, the null model provided the best account of choice data (Ef = 0.97, Ep > 0.99), discarding the possibility of a primacy bias that would be purely based on the option serial position. As control analyses, we

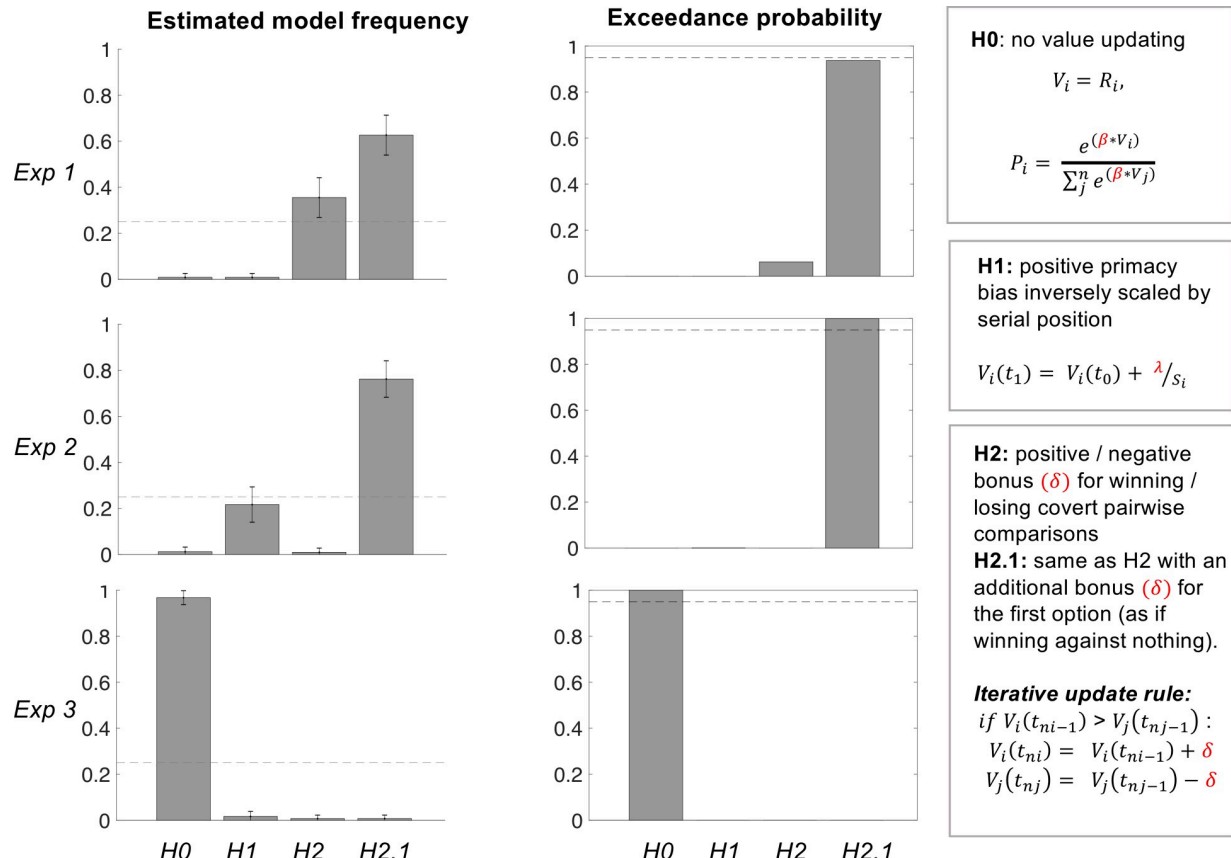

**Fig 4. Bayesian model comparison results.** All parameters have been fitted on choice data, separately for the three experiments. Models correspond to the different hypotheses (H0 to H2.1). The bias parameter λ in model H1 was bounded to be positive, in order to capture primacy effects. The bonus parameter δ in model H2 and H2.1 was not bounded, such that a positive posterior provides evidence for the existence of covert pairwise comparison. The recency bias model (H1 with negative parameter) and the pruning model (H3) were not included because they predicted a qualitatively opposite trend (increased probability of selecting the best option when presented later in the sequence), compared to what was observed in choice data. Exceedance probability is the likelihood that the considered model is more represented than the others, in the population from which participants were recruited. Dash lines represent chance level for expected frequency (0.25 because there are four models) and significance level for exceedance probability (0.95 because of the standard statistical criterion to reject random distributions).

included the pruning model (H3) and the recency model (same as primacy model but with a bias parameter bounded to be negative) in the model space. Results of the Bayesian model comparison were unchanged (see S5 and S6 Figs), making the conclusions robust to variations in the model space. We also checked that models such as H3, predicting a trend opposite to that observed in the data, was perfectly identifiable (at a 100% rate) from all the others in the recovery analysis.

To confirm that covert pairwise comparison was improving the fit of choice data, we tested the mean of posterior estimates obtained for the bonus parameter δ with the winning model (H2.1, assuming that the same bonus is applied to the first option and to winners of covert comparisons). When fitting model H2.1, the parameter δ was not constrained to be positive, so its sign can be interpreted as evidence for a bonus added to the winning option (and subtracted from the losing option). The posterior means (± SEM) for δ were 0.59 (± 0.089) in Exp 1 and 0.29 (± 0.11) in Exp 2. They were both significantly positive (Exp 1: $t(28) = 6.62$, p <0.001; Exp 2: $t(26) = 2.75$, p = 0.011), confirming that the model was applying a positive

bonus to options winning covert pairwise comparisons, and a negative bonus to losing options. The values of all fitted parameters are listed in S1 Table.

We then extended our computational analyses to assess whether model H2.1 would remain the most plausible compared to other variants.

First, we compared model H2.1 to a model in which the best option simply captures attention, receiving a bonus related to its saliency, as was proposed for salient attributes in choices between lotteries [30,31]. This saliency model was equivalent to H0 except for the bonus $\gamma$ attributed to the best option:

$$V_{best} = V_{best}.(1 + \gamma) \tag{7}$$

This new model tests the possibility that the bonus may be independent from the serial position of the best option, and therefore cannot explain the link between best-option choice rate and serial position observed in our data. Expectedly, model H2.1 won the comparison with the saliency model in both Exp 1 (Ef = 0.58, Ep = 0.81) and Exp 2 (Ef = 0.98, Ep > 0.99). This confirms that the order of sequential sampling matters for the probability of choosing the best option.

Second, we compared model H2.1 to a model H1.1 where the first item gained an extra bonus $\lambda_1$, relative to all subsequent items:

$$V_i(t_1) = V_i(t_0) + \lambda_1 + \frac{\lambda}{s_i} \text{ if } S_i = 1 \text{ and } V_i(t_1) = V_i(t_0) + \frac{\lambda}{s_i} \text{ if } S_i > 1 \tag{8}$$

The idea was to reduce the difference between models H1 and H2.1 to the critical impact of covert pairwise comparison. This model H1.1 would not produce a linear relationship between best-option choice rate and serial position, but simply accentuate its convexity. Expectedly, model H2.1 won the comparison with model H1.1 in both Exp 1(Ef = 0.96, Ep > 0.99) and Exp 2 (Ef = 0.62, Ep = 0.91). Because H1.1 might have been penalized for having one more parameter, we also tried a variant of H1.1 with λ1 replaced by λ (so the bonus for the first option is now 2λ), but the model comparison yielded similar results.

Third, we compared H2.1 to a model that we call probabilistic H2.1, because the outcome of covert pairwise comparison is probabilistically determined by a second softmax function, with a second inverse temperature parameter $\beta_c$ (different from the β parameter used in the softmax function that generates the probabilities of overt choices). This probabilistic version of H2.1 would be more coherent, as there is no reason to assume that covert choice is deterministic (meaning that the best option is always winning the comparison), while overt choice is known to be probabilistic. This makes the computations much heavier, as there are many paths in the tree of possible combinations (8, 16 and 32 for 4-, 5- and 6-item trials, respectively). For each possible path, values are updated in the same manner as in the deterministic model H2.1, and passed through the same final softmax function to generate choice probabilities. Then, for every option, the overall selection probability is obtained by summing the product of path and choice probabilities over all possible paths. These computations are detailed and illustrated in S1 Methods.

The deterministic version of H2.1 won the comparison with the new probabilistic one, in both Exp 1 (Ef = 0.87, Ep > 0.99) and Exp 2 (Ef = 0.92, Ep > 0.99). The fitted inverse temperature parameters suggest that indeed, covert choices were more deterministic than overt choices (Exp 1: $\beta_c = 0.24 \pm 0.047$, $\beta = 0.082 \pm 0.0065$; Exp 2: $\beta_c = 0.22 \pm 0.049$, $\beta = 0.10 \pm 0.009$). However, the bonus parameters were in the same range as with the deterministic model H2.1, and significantly different from zero in both experiments (Exp 1: $\delta = 0.57 \pm 0.13$; Exp 2: $\delta = 0.41 \pm 0.14$). Thus, even if using a softmax function for covert choice is theoretically more

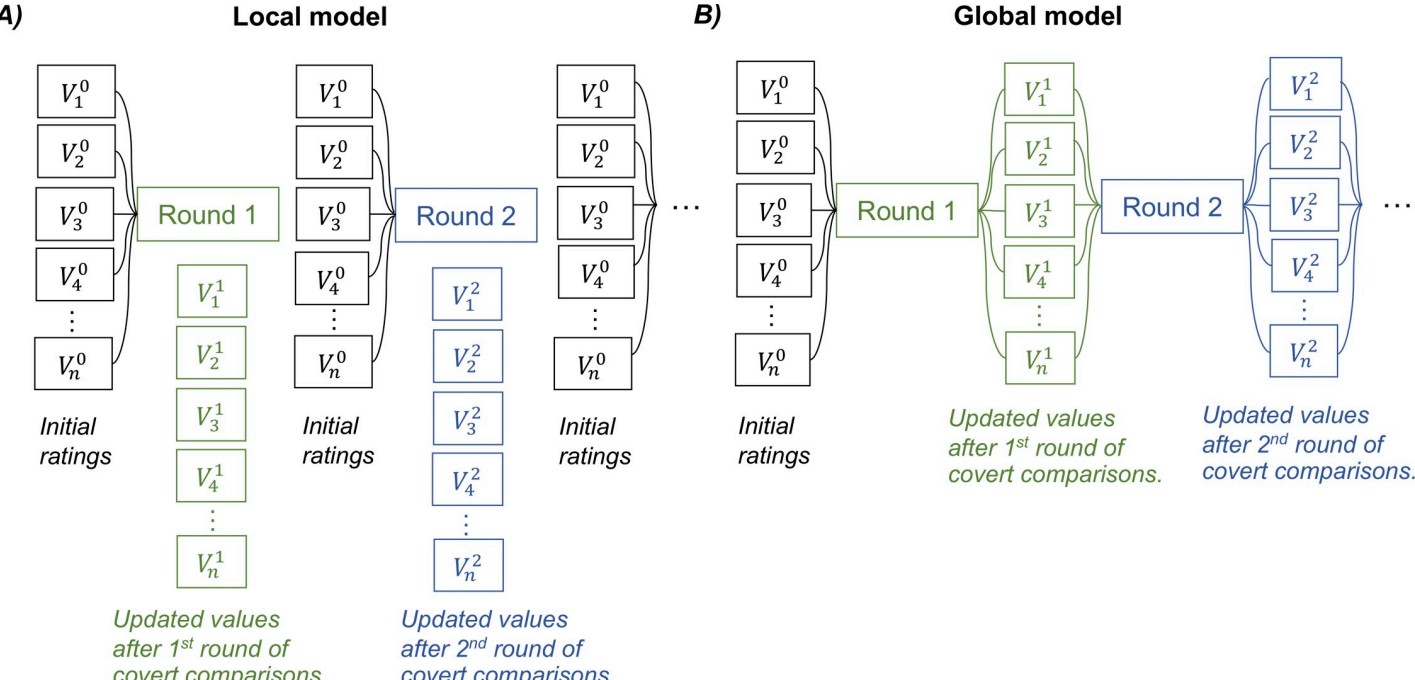

**Fig 5. Graphical illustration of local versus global effects of value updating following covert comparisons.** The initial values of item 1 to n are indicated as a $V_1^0$ to $V_n^0$ vector. Each item was presented several times during the choice task, each time paired with a different set of other items. After the first round, each item value is updated according to the covert pairwise comparison process (model H2.1), generating the $V_1^1$ to $V_n^1$ vector. Likewise, after the second round, the vector of item values is updated to $V_1^2$ to $V_n^2$. (A) The local model presumes that the updated values after covert comparisons are not stable and cannot carry over to subsequent trials. Thus, when a later trial presents the same item, its value has reversed back to the initial value (i.e., the likeability rating). (B) The global model assumes that the updated values are stable and thus can be passed on from one round to the next. The updated value from a previous trial can serve as a starting point for a new update in the next trial presenting the same item.

grounded, our choice dataset was not sensitive enough to benefit from this additional parameter. In any case, the impact of covert pairwise comparisons (the bonus added to the winning option and subtracted from the losing option) was significant in both the deterministic and probabilistic version of model H2.1.

**Across-trial effects.** Lastly, we assessed whether the shift in preference, captured with the bonus parameter in model H2.1, was confined to the ongoing trial or extended throughout subsequent trials of the experiment. We used the repetition of options, which were presented several times with different groupings in our design, to compare between a local model, where all trials start with initial ratings, and a global model, where trials start with updated ratings (see illustration in Fig 5). Thus, in the global model, the positive and negative bonus applied to winning and losing options is carried over to subsequent trials. Bayesian model comparison suggested that local model H2.1 provided a better account of choice data than global model H2.1 in both Exp 1 (Ef = 0.96, Ep > 0.99) and Exp 2 (Ef = 0.98, Ep > 0.99).

However, cognitive dissonance studies have suggested that choice-induced preference shift can last for quite a long time (18). We therefore assumed that, on top of the bonus gained through covert pairwise comparisons, there might be another bonus assigned to the option that was overtly chosen at the end of the trial. To assess this possibility, we developed a two-level H2.1 model, with two distinct bonus parameters, one due to covert comparisons and applied locally to the ongoing trial, and one due to overt choice and applied globally to all subsequent trials. Bayesian model comparison with local model H2.1 showed that two-level model H2.1 provided a better account of choice data in both Exp 1 (Ef = 0.98, Ep > 0.99) and Exp 2

(Ef = 0.98, Ep > 0.99). For completeness, we have included local H2.1, global H2.1 and two-level H2.1 in a single Bayesian model comparison performed in each experiment, separately (Fig 6). Results confirmed that two-level H2.1 was the overall best model in all experiments (Exp 1: Ef = 0.98, Ep > 0.99; Exp 2: Ef = 0.98, Ep > 0.99; Exp 3: Ef = 0.98, Ep > 0.99).

Note that in Exp 3, the fitted parameter $\delta$ for covert comparison was close to zero. Consistently, the two-level H2.1 model was outperformed by a hybrid model having just one bonus parameter for updating value following overt choice (Ef = 0.84, Ep > 0.99). This hybrid model also outperformed the global H0 model (Ef = 0.99, Ep > 0.99), confirming the bonus arising from overt choice, even when there was no bonus for covert choice. In the two other experiments (Exp 1 and Exp 2), where two-level model H2.1 was the winner, the posterior mean of the covert bonus parameter was much smaller than the overt bonus parameter (mean ± SEM: 0.48 ± 0.11/ 0.35 ± 0.13 compared to 24.73 ± 0.82 / 14. 08 ± 0.63 in Exp 1 / 2). Thus, although it provides a proof of concept for the process of covert pairwise comparison, the shift of preference induced during option sampling was less substantial than that induced by choice itself.

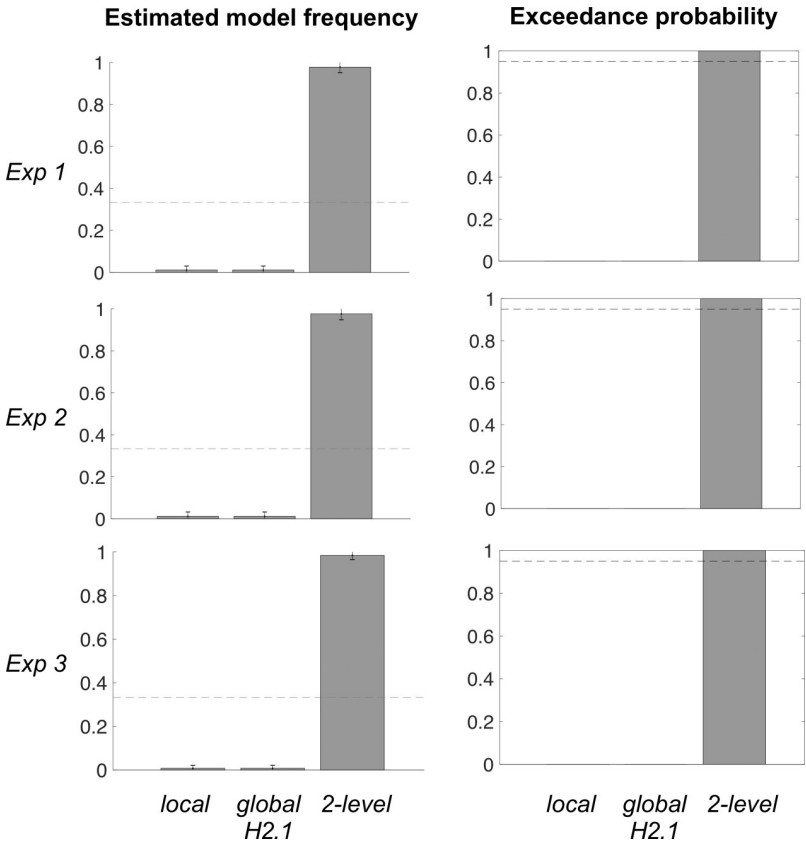

**Fig 6. Bayesian model comparison between variants of model H2.1.** All parameters have been fitted on choice data, separately for the three experiments. The model H2.1 implemented in previous comparisons is what we call here local model H2.1, which compared to two variants: the global model H2.1, in which value updates related to covert pairwise comparisons are carried over to all subsequent trials, and a two-level model H2.1, in which value updates induced by covert comparisons remain local, whereas value updates induced by overt comparisons (actual choices) are carried over to subsequent trials. Note the two-level model uses different bonus parameters for updating values after covert and overt comparisons. In the graph, gray dash lines represent chance level for expected frequency (0.33 because there are 3 models) and significance level for exceedance probability (0.95 because it corresponds to standard statistical criterion).

## Discussion

In this study, we suggested a choice model where the selection is prepared during sequential option sampling, through value updates induced by covert pairwise comparisons. Compared to a standard decision-making function, this model provided a better account of choice data collected in novel tasks where participants had to browse through a set of items before picking their favorite option. An important consequence is that the order in which options are sampled is not neutral: a good option presented early in the sequence will win more covert comparisons and hence will be more likely selected.

Alternative models would not make this prediction. In models typically used in decision neuroscience (softmax function of likeability ratings), designated here as the null model, choice is simply independent of serial positions. In memory accounts, choice only depends on serial positions, favoring either early or late options (according to primacy and recency effects). In our model, choice is influenced by an interaction between the option value and its position in the sequence: an early position is facilitating for good but not for bad options, and vice-versa for a late position. We also tested a variant of our covert pairwise comparison model, called the pruning model, where the worse options are just eliminated instead of being devalued. However, this model makes the opposite key prediction: an early position would be a disadvantage, even for a good option, because it would mean that it has to survive a high number of covert comparisons, which decrease its probability of being chosen, due to selection noise at every step.

This raises the question of why people implement these pairwise covert comparisons. The pruning model has a clear benefit: it would save space in working memory, as only one option identity, and not a set of option values, would be retained for the eventual choice. Indeed, in all models but the pruning model, choice probability is given at the end of sequential sampling by a softmax function of updated option values. Our suggestion is that spreading the alternatives, through the positive / negative bonus assigned to winning / losing options, increases the likelihood of selecting the best option and saves time when choice is to be made. A related consequence of covert value updating is a higher confidence (in making the right choice), which we argue is in fact what is maximized during decision-making [32]. This is trivial in perceptual decision-making, where one wants to be correct by instruction, and could apply as well in economic choice, where being correct means getting the best of alternative options, according to subjective preferences (24). The notion of confidence is tightly linked to choice response time, at both an empirical and theoretical level, the typical decision-maker being slower when less confident [33]. In economic choice, both confidence and response time depend on the distance in value between options, such that a spread of alternatives is beneficial [34]. Here, we have observed such trends: just as the likelihood of choosing the best option decreased with its serial position, confidence tended to decrease and response time to increase. However, our design was not optimized for these measures to be sensitive.

Another question pertains to the context in which the covert pairwise comparison process is implemented. Model predictions on choices were observed in Exp 1 and 2 but not in Exp 3, where options were all unmasked at the time of choice. Exp 3 can therefore be taken as a control for the impact of the imposed sequence of option sampling in Exp 1 and 2, because participants could visually resample all the options in an idiosyncratic order. Since we did not control for gaze direction, we could not keep track of their resampling pattern, which might have induced additional value updates, in accordance with our model. Another possibility is that participants did not implement the covert comparison process during sequential sampling in Exp 3—they just saved time by quickly browsing through options, and made their choice at the end, when all options were simultaneously unmasked. Exp 2 presents the interest of being

more ecological than Exp 1, as options were not made definitely invisible, but could only be resampled at the cost of losing time, as when surfing on the internet. In this case, we could track the order of option resampling, because participants had to click on the touch screen. When including the resampling process in our model, it still provided the best account of choice data. However, the resampling process was quite rare and thus, on average, the order of option sampling was not far from that programmed by design. Further experiments are needed to assess whether our computational account still holds in more natural settings, where the sampling sequence could be freely determined by subjects.

Although our covert pairwise comparison model was supported here by both model-free and model-based analyses, the evidence remains indirect. More direct evidence could be searched using neuroimaging techniques to track covert choices. Previous studies have shown that covert decisions to postpone a choice (during sequential sampling of information), which were inferred from computational analyses, could be uncovered from EEG signals [35]. Thus, it might be possible to decode neural representations of the current best option, such as its spatial location, either in a perceptual or in a motor format. Also, commitment to a choice (versus postponement), in the course of sequential sampling, has been related to decision-value signals recorded with fMRI, in brain regions such as the ventromedial prefrontal cortex [36]. Thus, it might be possible to assess the existence of value updates, following covert comparisons, by monitoring the activity of these brain regions. In this endeavor, our model might provide the computational probes needed to identify neural mechanisms operating covert choices.

Finally, we tested whether the eventual overt choice would also lead to value updating, as it has been repeatedly shown in studies using a rating-choice-rating paradigm [36]. We found that indeed, on top of the bonus arising from covert choice, the option that was overtly chosen received an additional bonus that was much larger and also more persistent. This additional effect of overt choice has been typically interpreted as resulting from the need to exhibit consistent behavior, possibly for social reputation management [37,38]. Although the consistency goal might have played a role for the effect of overt choice in our data, the fact that we also observed value updating for covert choice suggests the existence of a more implicit and automatic mechanism, as we previously argued [21]. Indeed, during debriefing interview, participants did not spontaneously report having adopted any strategy like iterative pairwise comparisons. Thus, action-induced spread of preference might encompass (at least) two sort of processes: an implicit one that is simply enhancing the contrast between option values, and an explicit one that intends to maintain apparent consistency. We note that in both cases, the additive model that we have been using, for the sake of simplicity, is unlikely to describe the true mechanisms producing the observed effects, since it provides no end to the increase in value every time an option is chosen. Unfortunately, a potential saturation in value updating could not be tested here, given the small number of presentations for a same option in our design.

In conclusion, we have provided evidence that sequential order of option sampling is not neutral for decision-making, which invites a reconsideration of standard decision theory. To this endeavor, we have suggested a first computational account that certainly needs to be refined in future research. The intention is of course not to suggest ways to manipulate choices, but to uncover systematic biases that may implicitly shape our preferences. Yet, this bias could be used in public policy to nudge people into choosing the option they prefer, by making sure it is encountered early when browsing across the available alternatives.

## Supporting information

**S1 Methods. Supplementary methods.**
(DOCX)

**S1 Fig. Simulations of choice data (under the settings of Exp 2 & 3).** Graphs show simulated probability of choosing the best option under the experimental setup of Exp 2 and 3, depending on the serial position of the best option in the sampling sequence (x-axis). Each of the plots is an average over 200 simulated datasets of 30 subjects implementing the corresponding model, for various (color-coded) number of options. Shaded areas indicate the average inter-participant SEM, across all datasets. Values of $\lambda$ and $\delta$ are indicated on the plots. The inverse temperature parameter was fixed to $\beta = 0.10$, which corresponds to the posterior estimates of the best model (H2.1) fitted to choice data in Exp 2. The simulations show that predictions about the link between P(best) and its serial position are similar to those made under Exp 1 settings. In particular, only H1 (with a positive bias) and H2 (including H2.1) predict a decreased choice rate when the best option is presented later in the sequence.
(TIF)

**S2 Fig. Comparison of behavioral results to model simulations in Exp 2 & 3.** (A) The upper graphs show the observed probability of choosing the best option, as a function of its serial position relative to that of the second-best option. A positive relative position means that the best was presented after the second-best option. Shaded areas indicate inter-participant SEM. Dotted lines show linear regression fit across all trials (with different numbers of options). Stars denote significance of t-test comparing regression slopes to zero. * $p < 0.05$, ** $p < 0.01$. (B) The bottom graphs show the simulated probability of choosing the best option, as a function of its serial position relative to that of the second-best option. Choice behavior in each condition was simulated using the best-fitting model with the posterior means for free parameters (see values of the inverse temperature $\beta$ and bonus $\delta$ indicated on the plots). Each of the plots is an average over 200 simulated datasets of 30 subjects implementing the corresponding model, for various (color-coded) number of options. Shaded areas indicate the average inter-participant SEM. across all datasets.
(TIF)

**S3 Fig. Model-free results about confidence and response time (Exp 1).** Graphs show the observed confidence rating (between 0 and 100) and response time (in seconds), as a function of the serial position of the best option (in Exp 1), for different (color-coded) number of options, averaged over trials in which the best option was chosen. Shaded areas indicate inter-participant SEM. Dotted lines show linear regression fit across trials (with different numbers of options). Star and circle denote significance or borderline significance of t-test comparing regression slopes to zero: * $p < 0.05$, ° $p < 0.1$.
(TIF)

**S4 Fig. Model recovery analysis (Exp 1).** Choice data have been simulated using the likeability ratings and the posterior parameters of participants in Exp 1. Recovery rate has been established on the basis of 50 simulations, each including a group of 30 random participants. Cells of the confusion matrix indicate the rate at which the model in row wins Bayesian comparison when the model in column was simulated. Bayesian comparison was applied to simulated data at the group level, between the four considered models, following the exact same procedure applied to observed data. Only winning models with exceedance probability > 0.95 have been included in the count. Note that when H2.1 model wins the comparison, the simulated model is either H2 or H2.1.
(TIF)

**S5 Fig. Bayesian model comparison results for an extended model space including the pruning model.** All parameters have been fitted on choice data, separately for the three experiments. Models correspond to the different hypotheses (H0 to H3). The H3 model in the plot is

the pruning model, which eliminates the option losing covert pairwise comparison at each step of the sampling process. Exceedance probability is the likelihood that the considered model is more represented than the others, in the population from which participants were recruited. Dash lines represent chance level for expected frequency (0.25 because there are four models) and significance level for exceedance probability (0.95 because of the standard statistical criterion to reject random distributions).
(TIF)

**S6 Fig. Bayesian model comparison results for an extended model space including the recency model.** All parameters have been fitted on choice data, separately for the three experiments. Models correspond to the different hypotheses (H0 to H2.1). The H1 primacy model in the plot is the same as in the main figure, with a bias parameter bounded to be positive, thus capturing primacy effects. The H1 recency model corresponds to model H1 with a bias parameter bounded to be negative. Exceedance probability is the likelihood that the considered model is more represented than the others, in the population from which participants were recruited. Dash lines represent chance level for expected frequency (0.25 because there are four models) and significance level for exceedance probability (0.95 because of the standard statistical criterion to reject random distributions).
(TIF)

**S1 Table. Fitted computational parameters (posterior mean ± SEM).**
(DOCX)

## Author Contributions

**Conceptualization:** Chen Hu, Philippe Domenech, Mathias Pessiglione.

**Data curation:** Chen Hu.

**Formal analysis:** Chen Hu.

**Funding acquisition:** Chen Hu, Mathias Pessiglione.

**Investigation:** Chen Hu.

**Methodology:** Chen Hu, Philippe Domenech, Mathias Pessiglione.

**Project administration:** Chen Hu, Mathias Pessiglione.

**Resources:** Chen Hu, Mathias Pessiglione.

**Software:** Chen Hu.

**Supervision:** Mathias Pessiglione.

**Validation:** Chen Hu, Philippe Domenech.

**Visualization:** Chen Hu, Mathias Pessiglione.

**Writing – original draft:** Chen Hu, Mathias Pessiglione.

**Writing – review & editing:** Chen Hu, Philippe Domenech, Mathias Pessiglione.

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
