## [Decision Letter · Decision Letter 0]

5 Jan 2020

Dear Dr Hu,

Thank you very much for submitting your manuscript 'Order matters: how covert value updating during sequential option sampling shapes economic preference' for review by PLOS Computational Biology. Your manuscript has been fully evaluated by the PLOS Computational Biology editorial team and in this case also by independent peer reviewers. The reviewers appreciated the attention to an important problem, but raised some substantial concerns about the manuscript as it currently stands. While your manuscript cannot be accepted in its present form, we are willing to consider a revised version in which the issues raised by the reviewers have been adequately addressed. We cannot, of course, promise publication at that time.

Sincerely,

Alireza Soltani

Associate Editor

PLOS Computational Biology

Samuel Gershman

Deputy Editor

PLOS Computational Biology

[LINK]

Reviewer's Responses to Questions

**Comments to the Authors:**

Reviewer #1: Summary

Chen at al. provides evidence that choice preferences are dynamically shaped during the option sampling process. Specifically, the order in which options are sampled during decision-making tasks can influence choices. Further, they show that the computational model in which option value is iteratively updated through pairwise comparisons provided the best account of participants' choice behavior. Together, these results suggest how the sequential option sampling process can affect economic decisions. The results provide an interesting extension of the existence decision-making mechanism and should be of interest to the field. However, the manuscript in its current form has several issues that need to be addressed before it would be suitable for publication. These issues and several major points for improvement are outlined below.

Major comments

1. It would be useful if the exposure time of each choice option (the time duration when each choice option was viewed) can also be taken into account in the analysis. Previous literature has shown that overt attention can enhance option value representation (Krajbich, et al, Nature Neuroscience, 2010; McGinty, et al, Neuron, 2016). Though the eye position signals in the current study were not recorded, it would be, nevertheless, worthwhile to investigate whether the exposure time of each choice option affects choice preferences.

2. More models need to be compared to solidify the results. For example, in Exp 3, H1 shows slightly better performance than H2. To better demonstrate the main results (Figure 4). It is useful to compare the performance between H1.1 and H2.1. Here, H1.1 is defined the same as H1 with an additional bonus for the first option. Besides, it is also not clear whether other forms of primacy model can better explain the behavior data. For example, V_i (t_1 )=V_i (t_0 )×(1+λ/s_i ).

Minor comments

1. To visualize the effect in Exp 3, it would be helpful to plot P(best) as a function of its absolute serial position, similar to Exp1.

2. In the last part of the results, the author shows that the two-level model H2.1 provided a better account than local model H2.1 in both Exp 1 and Exp 3. In order to better illustrate the overt choice effect in all experiment condition, it would be useful to test whether the two-level H0 model also best explain Exp2.

3. In Exp3, what were the options that participants resampled? Did resampling increase confidence?

4. The labels and the legends in Fig. 3 and Fig. S1 are mislocated or missing. Shade areas for S.E.M. are missing in Fig. 3 and Fig. S2.

5. Fig. S1 is not mentioned in the main text.

Reviewer #2: In their paper “Order matters: how covert value updating during sequential option sampling shapes economic preference”, Hu and colleagues investigate whether the order of presenting choice options in a multi-alternative choice task influences decisions. In three experiments, several variants of a task are tested, in which up to 6 choice options are presented sequentially. The central finding is that - contrary to standard economic theory - participants are more likely to choose the (subjectively) best option if this option occurs early in the sequence. Using computational modeling, the authors show that this effect is unlikely to be caused by a (memory-related) primacy effect. Instead, a model that assumes that options are compared in a covert pairwise manner and that the "winner" of this comparison receives a bonus to its subjective value explains the data best (at least under mnemonic constraints).

Overall, I am very enthusiastic about this paper which is well and succinctly written and addresses an important and timely research topic, which is the computational basis of seemingly irrational behavior in decisions between multiple (i.e., >2) alternatives. In many everyday life situations, such as grocery shopping, we are faced with a vast number of choice options, and it is unlikely that we can process all choice options at once. Thus, we are forced to search sequentially through the alternatives, and the experiments by Hu and colleagues create such situations in the lab. In my view, the combination of behavioral analyses and computational modeling shows in a convincing manner, that participants engage in covert pairwise comparisons and that these comparisons influence the final decision – a finding that may have substantial impact on research in various fields such as neuroscience, psychology and economics. Below are a few major and several minor suggestions for improving the manuscript.

Major comments:

1. As far as I understand, the “bonus” models (H2 and H2.1) assume that the option with the higher rating always wins the covert comparison and thus always receives the positive bonus. This may not be a very plausible assumption, given that we know that choices are always probabilistic (and that the model itself assumes a probabilistic choice process at the very end – see equation 1). Thus, it would be more plausible to assume that the probability that an option A wins the pairwise comparison against another option B depends on the value difference between A and B (quantified again by a softmax function, possibly with a different inverse temperature parameter than the one used in equation 1). Obviously, this makes things a bit more complicated because the winner of a pairwise comparison cannot be specified deterministically anymore. So, one has to take all potential paths of pairwise comparisons into account (plus the probability of these paths being realized) in order to specify the predictions of the model. Nevertheless, figuring this out should not be too difficult, and I would ask the authors to consider this model in addition to their existing set of models.

2. The authors basically propose the existence of covert decision processes on the basis of behavioral data (and computational modeling). For the Discussion, I think it would make a lot of sense to speculate about potential physiological or neuroscientific approaches to “directly” reveal these covert decision processes. Although this is clearly self-serving, I would like to point the authors to our work on uncovering the covert “decision not to decide” (Gluth et al., 2013, PLOS Comput Biol). This work together with related literature (e.g., O’Connell et al., 2012, Nat Neurosci) could form the basis for such a discussion point.

3. The order of experiments 1 to 3 appear to reflect the temporal order in which the experiments were conducted. Although this information is relevant (and should be mentioned at some point), I would still argue that it might be preferable to report experiment 2, which is kind of a control condition, either at the beginning or at the end. This would make the manuscript somewhat more accessible in my view.

Minor comments:

4. Author summary (p. 3): The very first sentence needs to be rephrased, because standard economic theory does NOT assume a two-step process of valuation and decision making. This is standard NEUROeconomic theory (e.g., the chapter by Glimcher in the 1st edition of the Neuroeconomics book). Traditional microeconomics does not say anything about how (or in how many steps) decisions emerge.

5. Introduction (p. 4): “…for which economic choice is construed as the selection of the option maximizing expected value”. The term “value” needs to be replaced by “utility”. Blaise Pascal assumed that people maximize expected value, but since Daniel Bernoulli, this has been dismissed and replaced by expected utility. In general, I don’t like that the term “value” is used so often throughout the manuscript (“utility” or “subjective value” are preferable).

6. Methods (p. 7): The number of participants (about 30 per experiment) appear to be sufficient for the purpose of the current study, but it should be noted how these numbers were determined (e.g., by a formal power analysis or by following previous studies).

7. Methods (p. 8): “Should two items have the same rating, we would compare the corresponding response time and assign a higher ranking to the one with a quicker response time”. I think this “rule of thumb” only makes sense for high-value items, but the opposite should be true for low-value items. For example, let’s say someone gives two options, A and B, a rating of “not at all” – but much quicker for A than for B. I would infer from this that the person is very sure that s/he doesn’t like A at all but is less sure in case of B. So, my prediction for a decision between A and B would be that s/he would choose B.

8. Methods (p. 12): w.r.t. equation 1, V_j is specified as “values of the o other options”. This is incorrect, because o must also include option i. Please clarify.

9. Methods (p. 15): It would be important to know whether the free parameters were restricted to any specific ranges (esp. for the bonus parameter delta).

10. Results (p. 18): “This suggests that using model H2.1 is adaptive in the sense that it makes decision-making closer to optimal policy on average”. I think this needs to be rephrased, because theoretically it is just wrong to make this claim (model H2.1 is less optimal than H0). I suggest to avoid the term “closer to optimal” and simply say that the pairwise covert comparisons are adaptive as they help participants to deal with the cognitive demands of the task.

Signed,

Sebastian Gluth

Reviewer #3: In this paper Hu and colleagues suggest that in contrast to the traditional view in economic decision theory, during sequential option sampling people compare every new alternative to the current best. To capture this idea, authors collected data from humans performing three variants of a novel multi-alternative decision task. Authors showed that in the cases were options were provided sequentially but masked prior to presentation of a new option, subjects’ choice behavior was best captured by a model in which every new alternative is compared with the current best. On the other hand, in the case where options were all unmasked at the time of choice, subjects’ choice behavior did not show any effect of the sequence order.

This is a well-designed experiment with interesting results which is of importance to our understanding of sequential decision making. However, I have number of major concerns that should be addressed to strengthen the conclusions that are drawn in the paper. Please find my concerns below:

Major issues:

The observed behaviors can be also put into the context of attentional weighting of outcomes under risk, which has been the focus of many studies in different contexts ([1-4]). How are H2(.1) & H3 models different than these suggested models? A detailed (model-free and model-based) comparison of these models is necessary to demonstrate the novelty of authors’ work.

It is puzzling to me why subjects’ continuous likability ratings are transformed to ranks at the beginning of the experiment. I understand that ranking makes combining data from all the subjects easier. However, distances between rating are informative and affected when moving from continuous likability ratings to discrete ranks. Authors should provide evidence that transformation of non-uniform likability ratings does not affect their observed results to a large extent.

Is there a change in the slope of probability of choosing the best option as a function of position for different number of items (Figure 3A Experiment 1)? Please comment on this.

Authors provide behavioral evidence supporting and against three models in their model-free analysis and later use model-based analysis to confirm those predictions. However, a few models were tested on the choice data without any behavioral evidence. This begs the question whether these different models are well identifiable in the experiments. To test this, authors should simulate each model with a range of parameters, and fit the simulated behavior with each model, then show that each model fits better data simulated by the same model.

Minor issues:

Page 8, Stimuli and apparatus & Page 15, Statistical analyses: Please include the version of Matlab used for data analysis and stimuli presentation.

Page 8, Behavioral tasks: Please add information on the number of trials in description of Experiments 1 and 3.

Page 10, third paragraph: … the persistence of choice-induced effects on subjective value. value updating. Please remove “value updating” at the end of the paragraph.

Page 15, Figure 2: B) “Shaded areas indicate the average S.E.M. across all datasets.” No shaded area is visible, please fix this issue.

Page 16, Results:

- Please add effect size analysis to your results section.

- Please add a table with mean and std of the extracted parameters for different models.

Page 17, Figure 3:

- Please fix titles, legends, x/y labels.

- A) Are these probabilities calculated over all subjects? Can authors show a similar figure with average and S.E.M of these probabilities calculated separately for each subject?

- B) “Shaded areas indicate the average S.E.M. across all datasets.” No shaded area is visible, please fix this issue.

Page 20, Figure 4: While H2.1 seems to be most likely model to explain the data in Exp. 1, the Exceedance probability of this model does not surpass chance level. Please comment on this.

Page 20 & 21: Please mention what the value after the plus/minus symbols (±) refers to (e.g., ±std).

Page 28, Figure S1:

- Please fix titles, legends, x/y-axis labels.

- Please add information on the modeled hypothesis to each panel.

- “Shaded areas indicate the average S.E.M.” No shaded area is visible, please fix this issue.

Page 29, Figure S2:

- “Shaded areas indicate the average S.E.M.” No shaded area is visible, please fix this issue.

- x-axis labels are missing.

- Please add panel labels to the figure instead of using location (left/right) to refer to panels.

References:

[1]. Birnbaum, M. H., & Navarrete, J. B. (1998). Testing descriptive utility theories: Violations of stochastic dominance and cumulative independence. Journal of Risk and Uncertainty, 17(1), 49-79.

[2]. Bordalo, P., Gennaioli, N., & Shleifer, A. (2012). Salience theory of choice under risk. The Quarterly journal of economics, 127(3), 1243-1285.

[3]. Farashahi, S., Azab, H., Hayden, B., & Soltani, A. (2018). On the flexibility of basic risk attitudes in monkeys. Journal of Neuroscience, 38(18), 4383-4398.

[4]. Spitmaan, M., Chu, E., & Soltani, A. (2019). Salience-driven value construction for adaptive choice under risk. Journal of Neuroscience, 39(26), 5195-5209.

**Have all data underlying the figures and results presented in the manuscript been provided?**

Reviewer #1: Yes

Reviewer #2: No:

Reviewer #3: Yes

PLOS authors have the option to publish the peer review history of their article (what does this mean?). If published, this will include your full peer review and any attached files.

Reviewer #1: No

Reviewer #2: Yes: Sebastian Gluth

Reviewer #3: No

---

## [Decision Letter · Decision Letter 1]

30 Apr 2020

Dear Dr. Hu,

We are pleased to inform you that your manuscript 'Order matters: how covert value updating during sequential option sampling shapes economic preference' has been provisionally accepted for publication in PLOS Computational Biology.

**In addition, per journal policy, we require that all data underlying the findings described to be fully available, without restriction, and from the time of publication. Currently, the authors mention that "data will be available on the Github database upon acceptance of the manuscript, upon demand", but no link to Github is provided. **

Best regards,

Alireza Soltani

Associate Editor

PLOS Computational Biology

Samuel Gershman

Deputy Editor

PLOS Computational Biology

Reviewer's Responses to Questions

**Comments to the Authors:**

Reviewer #1: The authors have successfully addressed all of my concerns.

Reviewer #2: The authors have done a great job in addressing all of my comments on the previous version of the manuscript. I just have two very minor remarks on the response to my last comment, but in general I am in favor of publication.

About this last comment: I would still claim that at least in the context of the current task, which is a static one in the sense that learning (and thus exploration) does not apply, the pairwise-comparison model does not implement the optimal policy (in a very strict, econometric sense). Nevertheless, I think the authors found a good way to answer my comment. I only want to suggest to replace the term "maximization of expected value" by "maximization of subjective value" or "maximization of utility". Furthermore, there is a ")" missing right before the last sentence of that new section.

Signed

Sebastian Gluth

Reviewer #3: I thank the authors for responding my questions. I have no further comments.

**Have all data underlying the figures and results presented in the manuscript been provided?**

Reviewer #1: Yes

Reviewer #2: None

Reviewer #3: Yes

PLOS authors have the option to publish the peer review history of their article (what does this mean?). If published, this will include your full peer review and any attached files.

Reviewer #1: No

Reviewer #2: Yes: Sebastian Gluth

Reviewer #3: No

---

## [Editor Report · Acceptance letter]

23 Jul 2020

PCOMPBIOL-D-19-02126R1 

Order matters: how covert value updating during sequential option sampling shapes economic preference

Dear Dr Hu,

I am pleased to inform you that your manuscript has been formally accepted for publication in PLOS Computational Biology. Your manuscript is now with our production department and you will be notified of the publication date in due course.

With kind regards,

Laura Mallard
